# Diet Quality Is Associated with Cardiometabolic Outcomes in Survivors of Childhood Leukemia

**DOI:** 10.3390/nu12072137

**Published:** 2020-07-18

**Authors:** Sophie Bérard, Sophia Morel, Emma Teasdale, Nitin Shivappa, James R. Hebert, Caroline Laverdière, Daniel Sinnett, Emile Levy, Valérie Marcil

**Affiliations:** 1Research Centre, Sainte-Justine University Health Center, Department of Nutrition, Université de Montréal, Montreal, QC H3T 1C5, Canada; sophie.berard@umontreal.ca (S.B.); sophia.morel@umontreal.ca (S.M.); emma.teasdale@umontreal.ca (E.T.); emile.levy@recherche-ste-justine.qc.ca (E.L.); 2Institute of Nutrition and Functional Foods, Laval University, Quebec City, QC G1V 0A6, Canada; 3Cancer Prevention and Control Program, University of South Carolina, Columbia, SC 29208, USA; shivappa@mailbox.sc.edu (N.S.); jhebert@mailbox.sc.edu (J.R.H.); 4Department of Epidemiology and Biostatistics, Arnold School of Public Health, University of South Carolina, Columbia, SC 29208, USA; 5Research Centre, Sainte-Justine University Health Center, Department of Pediatrics, Université de Montréal, Montreal, QC H3T 1C5, Canada; caroline.laverdiere@umontreal.ca (C.L.); daniel.sinnett@umontreal.ca (D.S.)

**Keywords:** survivors, pediatrics, oncology, acute lymphoblastic leukemia, dietary scores, cardiometabolic complications, nutrition

## Abstract

There is little information about how diet influences the health of childhood acute lymphoblastic leukemia (cALL) survivors. This study explores the associations between diet quality indices, cardiometabolic health indicators and inflammatory biomarkers among cALL survivors. Participants were part of the PETALE study (*n* = 241, median age: 21.7 years). Adherence to 6 dietary scores and caloric intake from ultra-processed foods were calculated. Multivariate logistirac regressions, Student *t*-tests and Mann-Whitney tests were performed. We found that 88% of adults and 46% of children adhered poorly to the Mediterranean diet, 36.9% had poor adherence to the World Health Organisation (WHO) recommendations and 76.3% had a diet to be improved according to the HEI-2015 score. On average, ultra-processed foods accounted for 51% of total energy intake. Low HDL-C was associated with a more inflammatory diet (E-DIITM score) and higher intake of ultra-processed foods. A greater E-DII score was associated with elevated insulin resistance (HOMA-IR), and consumption of ultra-processed foods was correlated with high triglycerides. Circulating levels of TNF-α, adiponectin and IL-6 were influenced by diet quality indices, while CRP and leptin were not. In conclusion, survivors of cALL have poor adherence to dietary recommendations, adversely affecting their cardiometabolic health.

## 1. Introduction

Pediatric cancers are the leading causes of disease-related mortality in Canadian children [1,2] and acute lymphoblastic leukemia (ALL) is the most common of these cancers [3]. An average of 880 Canadian children under the age of 15 are diagnosed with cancer each year and the incidence of pediatric cancers has remained relatively stable since 1992 [2]. Leukemia accounts for 32% of all pediatric cancers [2]. In recent years, therapeutics advances have achieved a survival rate for childhood ALL (cALL) that exceeds 90% [4]. Therefore, the population of survivors is growing in Canada and other parts of the world [2,5,6]. Unfortunately, cALL survivors are at greater risk of developing long-term chronic health complications [7]. For example, they are seven times more likely than their siblings to develop cardiac complications such as heart failure and myocardial infarction [8]. The PETALE study (*Prévenir les effets tardifs des traitements de la leucémie lymphoblastique chez l’enfant*), carried out in Sainte-Justine University Health Center (SJUHC) in Montreal, PQ, Canada showed that more than 60% of children and young adult survivors of cALL exhibited at least one cardiometabolic risk factor, including dyslipidemia, pre-hypertension or hypertension, obesity and insulin resistance [9]. Compared to participants of the same age in the general population, they were at higher risk of developing the metabolic syndrome [9]. Studies in other populations of cALL survivors displayed similar findings [10,11].

Several mechanisms have been proposed to explain the cardiometabolic side effects on cALL survivors [11]. Side effects of radiotherapy [12,13] and chemotherapy [14,15], gut microbiota modifications [16,17,18], poor eating habits [19,20] and physical inactivity [21,22] have all been pinpointed in this group. In the general population, it has been established that a healthy diet influences cardiometabolic risk [23,24] while poor adherence to dietary recommendations is negatively associated with cardiometabolic health outcomes [25,26]. To assess such associations, different validated food quality indices have been used [25,27,28,29]. In particular, food quality indices or nutritional scores are tools that provide an overall rating, on a numeric scale, indicating adherence level to nutrition recommendations or overall quality of diet [30,31].

Given the major health issues that face cALL survivors, it would be prudent for them to adopt healthy eating habits. However, some studies have reported their poor adherence to dietary guidelines [32,33,34]. Apparently, survivors do not eat enough fruits and vegetables [33,35,36], dairy products [35,36], whole grains [36,37], calcium and vitamin D [37] and they consume too much sodium, meat and processed foods [33]. Nonetheless, there is little information about how their eating habits may influence their cardiometabolic status.

Here, we studied diet quality of 241 cALL survivors of the PETALE cohort using nutritional scores including five using *a priori* dietary patterns (MEDAS, KIDMED, HDI-2018, HEI-2015, E-DII) and two *a posteriori* patterns (FRAP score and contribution of ultra-processed foods to total energy intake based on the NOVA classification). We also studied the associations between diet quality indices and participants’ cardiometabolic outcomes, as well as metabolic and inflammatory profile.

## 2. Materials and Methods

### 2.1. Study Population

Participants were recruited between January 2013 and December 2016 as part of the PETALE study at SJUHC in Montreal. A detailed presentation of the PETALE study and cohort is provided in Marcoux et al. [38]. The PETALE study was designed to characterize early-onset late adverse effects namely cardiometabolic complications, cardiotoxicity, neurocognitive problems, bone morbidity and quality of life in children, adolescents and young adults who had survived cALL [38]. Participants (*n* = 246) were of European-descent living in the Province of Quebec and were treated for ALL at SJUHC according to Dana-Farber Cancer Institution-ALL protocols 87-01 to 2005-01. Event-free patients who had not suffered from refractory ALL, relapsed or received hematopoietic stem cell transplant 5 years or more after diagnosis were eligible. Information on participants’ treatment protocols is provided in Levy et al. [9]. The Institutional Review Board of SJHUC approved the study (approval number 2013-479) and investigations were carried out in accordance with the principles of the Declaration of Helsinki. Written informed consent was obtained from study participants and/or parents/guardians.

### 2.2. Anthropometric, Clinical and Biochemical Assessment

Anthropometric, clinical and biochemical assessments were achieved, as previously described [9]. Waist-to-height ratio [WHtR, waist circumference (WC) (cm)/height (cm)] and body mass index [BMI, weight (kg)/height (m)^2^] were calculated [38]. Hypertension and pre-hypertension were determined according to current recommendations in adults and children [9]. Fasting insulin (pmol/L), glucose (mmol/L), total cholesterol (TC, mmol/L), triglycerides (mmol/L), low-density lipoprotein-cholesterol (LDL-C, mmol/L), high-density lipoprotein-cholesterol (HDL-C, mmol/L), apolipoproteins A-1 (Apo A-1, mg/mL) and B-100 (Apo B-100, g/L) and low-grade and visceral inflammation [C-reactive protein (CRP, μg/mL), tumor necrosis factor (TNF, pg/mL)-α, interleukin (IL, pg/mL)-6, adiponectin (ng/mL) and leptin (ng/mL)] were measured on fasting blood, as described previously [9,38]. The homeostasis model assessment of insulin resistance (HOMA-IR) was calculated [39]. Cardiometabolic complications were defined and categorized [9]—obesity (yes or no), pre-hypertension/hypertension (yes or no), insulin resistance (yes or no) and dyslipidemia (yes or no). Participants with ≥2 cardiometabolic complications were identified. Cut-off values and definitions used to determine cardiometabolic outcomes in children (<18 years old) and adults are described in Appendix A and were assessed as reported in former investigations [40,41,42].

### 2.3. Data Collection and Analysis for Dietary Intake

Participants’ dietary intake was recorded using a validated food frequency questionnaire consisting of 190 items. Data collection method, nutrient calculation and further classification in food groups and subgroups have been previously described [43]. For 191 participants, 3-day food records also were available and were analyzed with the Nutrific^®^ software developed by the Department of Food Science and Nutrition, Université Laval. Nutrient values from the Nutrific application were extracted from the 2010 Canadian Nutrient File.

### 2.4. Assessment of Diet Quality

Data collected with the food frequency questionnaires were used to calculate the dietary scores—Mediterranean Diet Adherence Screener (MEDAS), Mediterranean Diet Quality Index for children and adolescents (KIDMED), Healthy Diet Indicator (HDI-2018), Healthy Eating Index (HEI-2015), Energy-adjusted Dietary Inflammatory Index (E-DII^TM^) and Ferric reducing ability of plasma (FRAP). 3-day foods records were used for the NOVA Classification.

#### 2.4.1. Mediterranean Diet Adherence

For adults, adherence to the Mediterranean diet (MD) was determined by the 14-point MEDAS score. This score consists of 12 questions on food frequency and 2 questions on MD dietary habits—consumption of olive oil for cooking and >60 mL/day, vegetables (>4 portions/day), fruits (3/day), red meat (<1 portion/day), butter, margarine or cream (≤15 mL/day), sugary drinks (<355 mL/day), wine (≥3 glasses/week), legumes (≥3 portions/week), fish or seafood (≥3 portions/week), baked goods (<2/weeks), nuts (≥3 portions/week), preferring chicken/turkey instead of red meat and consuming boiled vegetables, pasta, rice or other dishes with a sauce of tomato, garlic or onions sautéed in olive oil. The questions are scored as either 0 or 1, reaching a maximum score of 14. Adherence is categorized as follows—≥10, strong adherence; 6–9, moderate adherence; ≤5, low adherence [44]. For children, MD adherence was assessed using an updated version of the KIDMED Index [45,46]. This index is based on 16 yes or no questions, with a total score ranging from—4 to 12. Scoring of +1 is attributed in accordance to the MD, while questions that do not reflect this type of diet are scored −1. Diet is classified in three levels—≥8, optimal MD; 4–7, improvement needed; ≤3, very low diet quality [45].

#### 2.4.2. Healthy Diet Indicator

The HDI is based on the 2003 World Health Organization (WHO) guidelines for prevention of chronic diseases [47]. The HDI-2018 was further developed according to the updated version of the WHO Fact Sheet 2018 [48]. It includes 9 components—≥400 g of fruits and vegetables per day, <30% of fat from total energy, <10% of saturated fat from total energy, 6–11% of polyunsaturated fatty acids from total energy, <10% of free sugars from total energy, ≥25 g of dietary fiber per day, ≥3500 mg of potassium per day, <2 g of sodium and <1% of trans fat from total energy. A value of +1 is assigned when intake is within the recommended range, otherwise the score is 0. The sum of the 9 components generates a score from 0 to 9. The level of adherence is determined as follows—≥7, strong adherence; 4–6, modest adherence; and ≤3, low adherence [49].

#### 2.4.3. Healthy Eating Index

The HEI-2015 assesses adherence to the Dietary Guidelines for Americans [50,51]. This score takes into account 14 dietary components expressed in 1000 kcal (total fruits, whole fruits, total vegetables, greens and beans, whole grains, dairy, total protein foods, seafood and plant proteins, refined grains, added sugars, fatty acids, sodium and saturated fats). Briefly, each healthy food component accounts for 5 or 10 points and unhealthy components for 0 point (i.e., refined grains ≥4.3 oz eq/1000 kcal; added sugar ≥26% of total energy; sodium ≥2 g/1000 kcal; saturated fatty acids—≥16% of total energy). The sum of the components leads to a score ranging from 0 to 100 [51,52]. A HEI-2015 score >80 indicates a good quality diet, a score ranging from 51–80 indicates a diet that needs improvement and a score <51 reflects poor quality diet [53].

#### 2.4.4. Energy-Adjusted-Dietary Inflammatory Index

The Dietary Inflammatory Index (DII^®^) was developed to assess the inflammatory potential of the diet. The development and validation of the DII has been published [54,55]. Briefly, it reflects the relationship between 45 food parameters and six inflammatory biomarkers reported by 1943 research articles published through 2010. Food and nutrient consumption was first adjusted for total energy per 1000 calories. To avoid the arbitrary use of raw intake amounts, the energy-adjusted dietary intake was standardized to an energy-adjusted worldwide representative diet database from eleven populations around the world, which was then multiplied by the literature-derived inflammatory effect score for each DII component and summed across all components to obtain the overall energy-adjusted score (E-DII^TM^). E-DII values can range from −8.87 (very anti-inflammatory diet) to +7.98 (very pro-inflammatory diet) [54]. For our study, we had access to 31 food parameters—alcohol, vitamin B_12_, vitamin B_6_, ß-carotene, caffeine, carbohydrates, cholesterol, energy, total fat, fiber, folic acid, iron, magnesium, monounsaturated fats, niacin, *n*-3 fatty acid, *n*-6 fatty acid, onion, protein, polyunsaturated fats, riboflavin, saturated fats, selenium, thiamin, trans fat, vitamin A, vitamin C, vitamin D, vitamin E, zinc, green/black tea.

#### 2.4.5. Ferric Reducing Ability of Plasma Score

The dietary antioxidant capacity was determined using the Antioxidant Food Table that measures the antioxidant content of over 3100 types of foods and beverages using a Ferric Reducing Ability of Plasma (FRAP) assay [56]. The FRAP assay determines the antioxidant capacity of individual food items to reduce ferric iron (Fe^3+^) to ferrous iron (Fe^2+^) [57] and has been widely used in nutritional science [58]. For each participant, a FRAP score was calculated, taking the FRAP value of every type of food derived from the Antioxidant Food Table (mmol/100 g) and multiplying it by the consumption frequency. These values were summed across all dietary sources of antioxidants, representing the total dietary antioxidant capacity.

#### 2.4.6. NOVA Classification

The NOVA classification (a name not an acronym) was used to assess the contribution of ultra-processed foods to the total dietary energy intake. This classification, developed by Monteiro and collaborators [59,60], categorizes foods and beverages according to their level of processing in four groups—Group 1, unprocessed or minimally processed foods (i.e., fresh fruits and vegetables, whole grains, legumes, eggs); Group 2, culinary ingredients (i.e., sugar, salt, oils); Group 3, processed foods (i.e., canned fruits, vegetables and legumes, dried, smoked, salted or sweetened meat, fish and nuts); Group 4, ultra-processed foods (i.e., soft drinks, sweet and salty snacks, commercial breads, cereals, flavored yogurts). For our study, foods and beverages were categorized by 3 registered dietitians. Caloric intake from ultra-processed foods (Group 4) was calculated as a percentage of the total daily energy intake.

### 2.5. Statistical Analysis

Distribution of variables was evaluated using the Kolmogorov-Smirnov test. Nutritional scores were classified into tertiles according to adherence levels. Crude and adjusted multivariate logistic regression models tested the associations between dietary scores and metabolic outcomes by calculating odds ratio and 95% confidence intervals. Potentially confounding variables including gender and survival time were incorporated in the analysis for all the scores. Energy intake was included as a confounding variable for the scores for which energy is not taken into account when calculated (MEDAS, KIDMED, HDI, FRAP, NOVA classification). For inflammatory biomarkers, differences in mean values between groups were tested using the Student t-test or Mann-Whitney test. *p* values < 0.05 were considered statistically significant. Statistical analysis was performed using SPSS (IBM SPSS Statistics for Windows, Version 25, Armonk, NY: IBM Corp.).

## 3. Results

### 3.1. Descriptive Statistics

Table 1 provides a summary of participants’ characteristics. One participant was excluded because he did not meet the inclusion criteria for PETALE and five participants were excluded due to missing nutritional data. A total of 241 participants were included in the study, comprised of 156 adults (≥18 years old) and 85 children (<18 years old). The proportion of males was 49.4%. The median age at visit was 21.3 years and median time since the end of treatment was 12.9 years. Overall, about 1/3rd (32.4%) of participants were obese, 41.1% had dyslipidemia, 16.6% had insulin resistance and 12% had pre-hypertension or hypertension and 29% had ≥2 cardiometabolic complications (cut-off values defined in Appendix A).

The descriptive statistics for dietary scores are summarized in Table 2 and their distributions are presented in Figure 1. According to the mean MEDAS and KIDMED scores, 88% of adults and 46% of children adhered poorly to the MD, whereas 12% of adults and 51% of children had moderate adherence. No adult and only 4% of children strongly adhered to the MD. Adherence to the WHO recommendations, qualified using the HDI-2018 score, showed that 89 participants (36.9%) had poor adherence, 136 (56.4%) modest adherence and 16 (6.6%) strong adherence. The HEI-2015 score revealed that 52 participants (21.6%) had poor quality diet, 184 (76.3%) had a diet to be improved and 5 (2.1%) had a diet of good quality. The E-DII scores ranged from −4.80 (most anti-inflammatory score) to +3.23 (most pro-inflammatory score). The FRAP score was between 1.96 mmol/day (lower total antioxidant capacity) and 48.8 mmol/day (higher total antioxidant capacity). On average, ultra-processed foods accounted for more than half of participants’ total energy intake, with a maximum of 93.4%.

### 3.2. Association between Dietary Scores and Anthropometric and Metabolic Parameters

Our results show that a more inflammatory diet, estimated with the E-DII score, was positively associated with high HOMA-IR (tertile 2 vs. 1: OR: 2.667, *p* = 0.03) and low HDL-C (tertiles 2 and 3 vs. 1: OR: 2.359, *p* = 0.02) (Table 3). High consumption of ultra-processed foods was positively correlated with having low HDL-C (tertile 3 vs. 1: OR: 3.885, *p* = 0.004) and high TG (tertile 2 and 3 vs. 1: OR: 4.021, *p* = 0.03) (Table 4). In adults, there were tendencies for a protective association between a higher MEDAS score and high WC (tertile 2 vs. 1: OR: 0.374, *p* = 0.059) and low HDL-C (tertile 2 vs. 1: OR: 0.401, *p* = 0.063); however, these results did not reach statistical significance (Appendix A). In children, a better adherence to the KIDMED was inversely associated with high SBP (tertiles 2 and 3 vs. 1: OR: 0.193, *p* = 0.05) (Appendix A). Finally, there was no association between adherence to the HDI-2018, HEI-2015 and FRAP scores and anthropometric and metabolic parameters (Appendix A).

Assessing the associations between diet and cardiometabolic complications revealed that a more pro-inflammatory diet was associated with an increased risk of having two or more risk factors (tertile 2 vs. 1: OR: 2.506, *p* = 0.01 and tertiles 2 and 3 vs. 1: OR: 2.076, *p* = 0.03) (Table 5). Similar results were observed when risk factors were analyzed individually: a more pro-inflammatory diet was positively associated with insulin resistance and hypertension and a higher consumption of ultra-processed foods was associated with dyslipidemia, but results were not statistically significant (Appendix A).

### 3.3. Association between Dietary Scores and Inflammatory Biomarkers

Participants who had a better adherence to the HEI-2015 score had lower level of TNF-α (tertile 3: 1.99 pg/mL vs. tertile 1: 2.28 pg/mL, *p* = 0.01), while levels were higher in those having a more inflammatory diet (tertile 3: 2.89 pg/mL vs. tertile 1: 2.10 pg/mL, *p* = 0.07), without reaching statistical significance (Appendix A). Next, the group was stratified according to participants’ obesity status (Figure 2). In non-obese participants, TNF-α levels were lower in those with higher HEI-2015 and KIDMED scores (tertile 3: 1.98 pg/mL vs. tertile 1: 2.23 pg/mL, *p* = 0.049 and tertile 2: 1.83 pg/mL vs. tertile 1: 2.52 pg/mL, *p* = 0.049, respectively). An inverse trend was observed with a greater proportion of ultra-processed foods (tertile 3: 2.36 pg/mL vs. tertile 1: 1.97 pg/mL, *p* = 0.06) (Figure 2). There was no effect of the scores in obese participants only.

Adults with a better adherence to the MD had higher levels of adiponectin (tertile 3: 19.41 ng/mL vs. tertile 1: 14.10 ng/mL, *p* = 0.004) (Appendix A). Similar differences according to the MEDAS score were found in both non-obese and obese adults but were not observed in children with the KIDMED score (Figure 2). In obese participants, adiponectin levels were lower with a pro-inflammatory diet (tertile 3: 11.44 ng/mL vs. tertile 1: 16.77 ng/mL, *p* = 0.044) and were higher with a better adherence to HDI-2018 scores (tertile 3: 15.76 ng/mL vs. tertile 1: 11.22 ng/mL, *p* = 0.044) (Figure 2).

Differences in IL-6 levels were observed only when obese and non-obese participants were analyzed separately (Appendix A). IL-6 levels were higher among obese and non-obese participants who consumed more ultra-processed foods (tertile 3: 0.83 pg/mL vs. tertile 2: 0.65 pg/mL, *p* = 0.04 and tertile 3: 0.77 pg/mL vs. tertile 2: 0.34 pg/mL, *p* = 0.004, respectively) (Appendix A). Also, non-obese participants who had a better adherence to the HDI-2018 score had lower IL-6 levels (tertile 3: 0.35 pg/mL vs. tertile 1: 0.55 pg/mL, *p* = 0.02) (Appendix A). No statistical differences were found in the levels of CRP and leptin according to dietary scores (data not shown).

## 4. Discussion

We found that survivors of cALL of the PETALE cohort adhered poorly to the MD, had a modest adherence to the WHO recommendations and, according to the HEI-2015 score, the quality of their diet had to be improved. On average, more than half of participants’ daily energy intake was provided by ultra-processed foods. We also found that adherence to dietary scores was associated with several cardiometabolic outcomes, namely HDL-C, triglycerides, HOMA-IR, blood pressure and accumulating cardiometabolic risk factors. Diet quality also influenced circulating TNF-α and adiponectin levels.

It is well documented that, in the general population, the adherence to dietary guidelines and the quality of diet are sub-optimal [61]. It appears that survivors of childhood cancer are no exception. Several studies have demonstrated that cALL survivors adhere poorly to dietary recommendations [32,33,34]. By using 7 validated dietary scores, our study complements these observations. The use of dietary scores assesses, through a holistic approach, both quality and variety of diet. In our population, a large proportion of total caloric intake was provided by ultra-processed foods and adherence to the MD and the WHO recommendations was generally poor. Our results are consistent with data observed in previous studies, showing that survivors consume a high intake of free sugars, processed and refined foods, sodium and few fruits and vegetables [32,33]. Given that childhood cancer survivors are at high risk of metabolic and cardiovascular diseases [9,14] and of secondary cancer [62], adherence to healthy dietary patterns should be promoted in this population.

In our study, a better adherence to the MEDAS score was inversely associated with the risk of having low HDL-C, whereas positive associations were found with a more pro-inflammatory diet (E-DII score) and higher intake of ultra-processed foods. Previously, our group has highlighted the influence of nutrient and food group intakes on the risk of having low HDL-C in the PETALE cohort [39]. In Brazilian adults without cancer, a Traditional dietary pattern, characterized by a high consumption of rice and legumes and a low consumption of red meat, fat and sugar was associated with higher HDL-C [27]. An inverse association was found between the E-DII score and HDL-C levels in an American adult population [63]. HDL particles play a protective role in cardiovascular health by their involvement in reverse cholesterol transport [64] and their anti-inflammatory [65], antioxidant [66] and anti-thrombotic properties [67]. Several dietary components can improve HDL function. For example, the polyphenols contained in olive oil can increase reverse cholesterol transport and particle size, promote better stability by reducing triglycerides content and improve redox status [68]. Also, high doses of eicosapentaenoic acid increase reverse cholesterol transport and improve HDL anti-inflammatory and antioxidant functions [69]. Based on our results and considering that cALL survivors are more likely to present with low HDL-C [70,71], the improvement of diet quality should become an important aspect of long-term care management.

We found that a greater contribution of ultra-processed foods to total energy intake was associated with having high triglycerides. While data on the relationship between consumption of ultra-processed foods and cardiometabolic outcomes are growing [72], their impact on triglyceride levels remains uncertain [73]. Nonetheless, circulating levels of triglycerides are an independent risk factor for cardiovascular disease [74]. Also, low HDL-C levels were associated with high levels of triglycerides in previous studies [75,76]. Compared with the general population, survivors are eight times more likely to die from cardiovascular-related disease [77], which argues for a reduction of ultra-processed food consumption.

Participants consuming a more pro-inflammatory diet, measured with the E-DII score, were at higher risk of having a high HOMA-IR, an indicator of insulin resistance. Diet is an important contributor in the development of type 2 diabetes and insulin resistance [78] and other inflammation-related diseases [79]. Dietary components can exert pro-inflammatory or anti-inflammatory effects, which may influence the risk of inflammatory diseases, among other atherosclerosis [79]. Previous studies have shown that the DII score is associated with increased inflammatory biomarkers [80], incidence of cardiovascular disease [81], some cancers [82,83] and adiposity measures [84]. In a cross-sectional study of 2975 Iranian adults, the association between the DII score and the risk of glucose intolerance disorders could not be demonstrated, although trends were found between a higher DII and fasting blood glucose and HOMA-IR [85]. Likewise, a Western diet typically rich in pro-inflammatory foods, has been associated with altered glucose metabolism, insulin resistance and cardiovascular diseases [86,87]. More studies are needed to clearly assess if and how a “pro-inflammatory diet” can contribute to the development of insulin resistance and type 2 diabetes, especially in populations of childhood cancer survivors who have been exposed to high levels of inflammation and oxidative stress during their treatments.

In our cohort, a better adherence to the KIDMED and HDI-2018 scores was negatively associated with the risk of having pre-hypertension/hypertension and a more inflammatory diet was associated with an increased risk. Similarly, in a cohort of Brazilian adults, there was an inverse association between a diet rich in fruits, vegetables, legumes, whole grains and fish and high systolic blood pressure [27]. However, no relation between blood pressure and the adherence to the MD was found in adolescents [88]. We previously demonstrated that participants of the PETALE cohort were at increased risk of pre-hypertension and hypertension compared to the general population (relative risk of 2.59) [9]. While the underlying mechanisms remain misunderstood, the exposure of the cardiovascular system at an early age to the toxic effects of chemotherapeutic agents and to radiotherapy could cause endothelial damage [89]. Clearly, finding nutritional avenues to alleviate this side effect in the long term would benefit this population.

Although we found trends for associations between several dietary scores and indicators of adiposity (body mass index (BMI) and waist circumference), statistical significance was never achieved, perhaps owing to our limited sample size. In the literature, such associations are inconsistent. In a 6-year longitudinal study, better adherence to a MD was associated with lower waist circumference [90] but another study in adult women found no association [91]. In children, good adherence to the KIDMED score resulted in a 30% decreased odds of being overweight or obese [88]. In a cohort of 2967 adults, consumption of ultra-processed foods was associated with in increased risk of overweight and obesity [92]. However, in children, there was no association between the HEI-2015 score and BMI [93]. The causes of obesity are complex and multifactorial. Not only can the quantitative and qualitative aspects of diet have an impact, but other factors such as physical activity [94], sleeping habits [95] and genetics [96], contribute to maintaining body weight via the regulation of metabolic and endocrine processes. Although it appears intuitive, the contribution of diet quality in the maintenance of body weight and adiposity after childhood cancer remains to be confirmed.

Abdominal obesity is recognized as a risk factor for cardiovascular disease and type 2 diabetes [97]. It is characterized by increased adipose tissue surrounding the intra-abdominal organs. As a hormonally active tissue, visceral adipose tissue releases different bioactive molecules and hormones, such as adiponectin, leptin, TNF-α and IL-6. In our cohort of cALL survivors, we found associations between dietary scores and the inflammatory biomarkers TNF-α, IL-6 and adiponectin, while none was found for leptin and CRP. TNF-α is associated with low-grade inflammation, possibly leading to insulin resistance and diabetes [98] and contributes to the pathogenesis of atherosclerosis [99]. In non-obese participants of our cohort, greater adherence to the HEI-2015 and KIDMED scores was associated with lower TNF-α levels. Conversely, there was a positive trend observed with eating a more inflammatory diet or ultra-processed foods. In parallel, no statistically significant difference was found in obese participants, although a similar trend was found for HEI-2015 adherence. In the literature, a higher E-DII score has been associated with greater TNF-α levels in adolescents [80]. In adults with type 2 diabetes or at least 3 risk factors (hypertension, dyslipidemia or a family history coronary heart disease), a nutritional intervention based on the MD led to lower plasma concentrations of TNF receptors [100]. We also found higher IL-6 levels in relation to ultra-processed foods in both obese and non-obese participants and lower IL-6 with higher HDI-2018 score in non-obese. IL-6 has been identified as a marker for metabolic disorders and cardiovascular disease, although its exact in vivo pathophysiological significance remains unknown [101]. Some literature also supports a link between circulating IL-6 and diet. In women, consumption of trans fatty acids was positively associated with IL-6 levels but only in participants with higher BMI [102]. In populations at high cardiometabolic risk, an MD supplemented with olive oil or nuts reduced IL-6 levels [103]. Additionally, adiponectin produced by the adipose tissue, is responsible for the modulation of several metabolic processes such as glucose hemostasis and fatty acid oxidation, in addition to having anti-inflammatory properties [104,105]. Circulating levels of adiponectin have been inversely associated with the metabolic syndrome [106,107]. In our cohort, participants with a better adherence to the MD had higher adiponectin levels and this was observed in both obese and non-obese adults. Additional differences were found only in obese participants, namely with the HDI-2018 and E-DII scores. Similarly, a literature review of 10 studies on dietary patterns and adiponectin levels concluded that adherents to healthy dietary patterns have higher levels of circulating adiponectin [108]. The reasons explaining why, in our study, different results were observed in obese and non-obese participants are unclear. One can speculate that, with greater adiposity, the contribution of diet to one’s inflammatory state could be more or less important, depending on the pathway. Nonetheless, the small sample size of the sub-group might have precluded reaching statistical significance.

In our study, no associations between diet quality and the biomarkers leptin and CRP were found. Leptin is an important modulator of food intake and energy balance. Circulating leptin is strongly associated with BMI and the degree of adiposity [109]. Leptin levels were found higher in female cALL survivors compared to controls, which was not observed in male survivors [110]. However, leptin levels were not associated with dietary habits in another cohort [111]. CRP is a biomarker of the inflammatory status and a predictor of cardiovascular events [112]. CRP was associated with trans fatty acids in obese women [102] and was reduced following an MD intervention [103]. While the literature on diet quality and biomarkers of inflammation is divergent, further studies are needed to elucidate the role of diet on inflammatory processes in survivors of cALL.

The PETALE cohort is mostly comprised of AYA cALL survivors, a population that is understudied and that faces particular challenges in terms of compliance with treatment [113,114] and ability to follow dietary recommendations. In particular, low consumption of fruits and vegetables [32,115], fiber [34] and high intakes of high-fat foods [115] have been reported in AYA cancer survivors. Besides, high intakes of refined grains [61,116], empty calories [116] fruit juice and food high in sugar and fat [61] were noted in children and adolescents from the general population. Therefore, it is of high interest to study dietary and lifestyle habits in the very specific population of AYA cancer survivors.

A strength of our study is the use of numerous dietary scores to evaluate the quality of diet and its association with cardiometabolic health in survivors of cALL. The use of a validated food frequency questionnaire tailored for our population and the thorough biochemical and clinical characterization of the PETALE cohort are other strengths. One important study limitation is the sample size, which reduce the likelihood of obtaining more statistically significant associations. Also, it is known that questionnaires are associated with social approval and desirability biases [117,118] and that self-assessment of diet makes measurement error inevitable. While the MD can be assessed by several scores and *a posteriori* analyses, only the MEDAS and KIDMED scores were used in our study. Moreover, given the exploratory nature of the study, *p* values were not adjusted for multiple testing. It is possible that other confounding variables were not measured or accounted for in the analysis. Validation of our findings in other cohorts of cALL survivors is necessary. The lack of a non-cancer control group is another study limitation that prevents comparison with the general population. Finally, our study findings identified associations and do not explain how the different components of diet can modulate cardiometabolic health outcomes.

## 5. Conclusions

Our study showed that adolescent and young adult survivors of cALL do not have good adherence to dietary guidelines. Our results support a beneficial role for a high-quality diet on survivors’ cardiometabolic health, reflected by adherence to the MD, the WHO recommendations, the American Dietary Guidelines, as well as by a limited consumption of pro-inflammatory and ultra-processed foods. A better understanding of how diet and dietary components can affect the health of cALL survivors will allow the development of tailored recommendations and nutritional strategies for this high-risk population.

## Figures and Tables

**Figure 1 nutrients-12-02137-f001:**
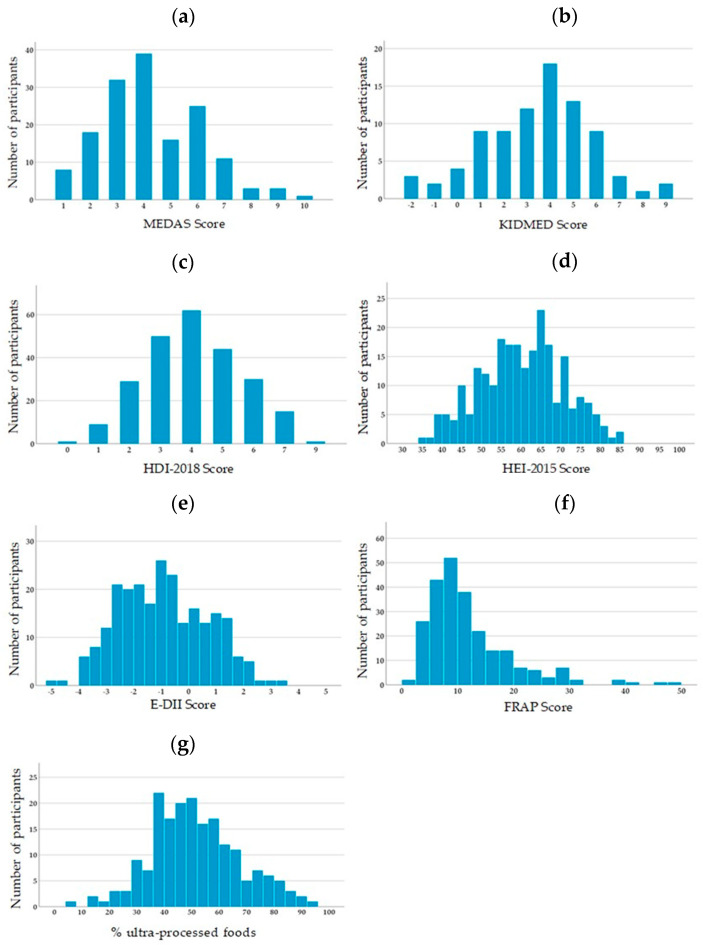
Distribution of the dietary scores among participants. (**a**) MEDAS (Mediterranean Diet Adherence Screener) score; (**b**) KIDMED (Mediterranean Diet Quality Index for children and adolescents) score; (**c**) HDI (Healthy Diet Indicator)-2018 score; (**d**) HEI (Healthy Eating Index)-2015 score; (**e**) E-DII (Energy-adjusted dietary Inflammatory Index) score; (**f**) FRAP (Ferric reducing ability of plasma) score; (**g**) contribution of ultra-processed foods according to the NOVA classification.

**Figure 2 nutrients-12-02137-f002:**
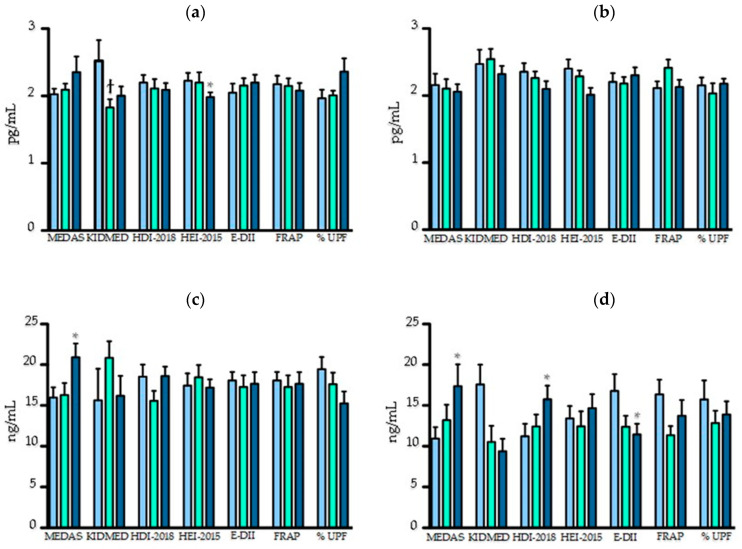
Inflammatory biomarker levels (*y* axis) according to nutritional scores (*x* axis). (**a**) TNF-α (pg/mL) in non-obese; (**b**) TNF-α in obese; (**c**) adiponectin (ng/mL) in non-obese; (**d**) adiponectin level in obese; * *p* < 0.05 tertile 3 vs. 1; ^ϯ^
*p* < 0.05 tertile 2 vs. 1. MEDAS, Mediterranean Diet Adherence Screener; KIDMED, Mediterranean Diet Quality Index for children and adolescents; HDI-2018, Healthy Diet Indicator; HEI-2015, Healthy Eating Index; E-DII, Energy-adjusted dietary Inflammatory Index; FRAP, Ferric reducing ability of plasma; Ultra-processed foods.

**Table 1 nutrients-12-02137-t001:** Demographic and clinical characteristics of participants.

	Total *N* = 241	Adults *N* = 156	Children *N* = 85
Median (range or interquartile range ^1^)			
Age at visit, years (range)	21.3 (8.5–40.9)	24.6 (18.0–40.9)	16.2 (8.5–17.9)
Age at cancer diagnosis, years (range)	4.7 (0.9–18.0)	6.5 (0.9–18.0)	3.5 (1.3–10.9)
Time since end of treatment, years (range)	12.9 (3.3–26.1)	16.11 (3.9–26.1)	9.6 (3.3–13.4)
Gender (males, %)	49.4	49.4	49.4
CRT exposure (*n*, %)	142 (58.9%)	108 (69.2%)	34 (40.0%)
BMI (kg/m^2^)	23.5 (20.9–26.1)	24.3 (21.7–27.4)	21.8 (19.2–24.1)
WHtR	0.50 (0.46–0.55)	0.51 (0.48–0.58)	0.49 (0.45–0.52)
WC (cm)	85.7 (76.0–93.0)	89.0 (79.5–97.0)	79.9 (72.0–84.0)
Glucose (mmol/L)	5.0 (4.8–5.3)	5.0 (4.8–5.4)	5.0 (4.8–5.3)
Insulin (pmol/L)	53.3 (37.6–75.8)	50.1 (35.2–68.6)	58.1 (42.9–81.1)
HOMA-IR	1.7 (1.2–2.5)	1.7 (1.1–2.3)	1.9 (1.3–2.7)
TC (mmol/L)	4.37 (3.87–5.01)	4.59 (4.10–5.15)	4.18 (3.61–4.63)
TG (mmol/L)	0.91 (0.66–1.25)	0.97 (0.72–1.38)	0.82 (0.62–1.07)
LDL-C (mmol/L)	2.57 (2.13–3.16)	2.73 (2.22–3.32)	2.36 (2.04–2.82)
HDL-C (mmol/L)	1.30 (1.12–1.49)	1.31 (1.13–1.52)	1.29 (1.09–1.45)
SBP (mmHg)	115 (108–124)	117 (110–125)	112 (104–119)
DBP (mmHg)	65 (59–70)	67 (63–72)	62 (57–65)
	*n* = 78	*n* = 38	*n* = 40
Apo A1 (mg/mL)	2.32 (1.94–2.65)	2.46 (1.98–2.72)	2.18 (1.90–2.52)
Apo B100 (g/L)	0.82 (0.71–0.94)	0.85 (0.72–0.97)	0.79 (0.68–0.92)

^1^ Interquartile range: 25th and 75th percentiles, unless otherwise specified. CRT, cranial radiation therapy; BMI, body mass index; WHtR, waist-to-height ratio; WC, waist circumference; HOMA-IR, homeostasis model assessment-insulin resistance; TC, total cholesterol; TG, triglyceride; LDL-C, low-density lipoprotein-cholesterol; HDL-C, high-density lipoprotein-cholesterol; SBP, systolic blood pressure; DBP, diastolic blood pressure.

**Table 2 nutrients-12-02137-t002:** Descriptive statistics of the dietary scores.

Score (Range)	*N*	Mean	Median	SD	Min	Max
MEDAS (0–14)	156	4.26	4.00	1.87	1.00	10.00
KIDMED (−4–12)	85	3.45	4.00	2.36	−2.00	9.00
HDI-2018 (0–9)	241	4.06	4.00	1.56	0.00	9.00
HEI-2015 (0–100)	241	59.78	60.00	10.58	35.00	85.00
E-DII (−8.87–7.98)	241	−0.92	−1.06	1.61	−4.80	+3.23
FRAP (≥0 mmol/day)	241	11.89	9.78	7.64	1.96	48.79
% UPF (0–100%) ^1^	191	51.33	50.69	16.11	6.93	93.36

^1^ Percentage of caloric intake from ultra-processed foods (Group 4) based on the NOVA classification. MEDAS, Mediterranean Diet Adherence Screener; KIDMED, Mediterranean Diet Quality Index for children and adolescents; HDI-2018, Healthy Diet Indicator; HEI-2015, Healthy Eating Index; E-DII, Energy-adjusted dietary Inflammatory Index; FRAP, Ferric reducing ability of plasma; UPF, Ultra-processed foods; SD, standard deviation; Min, minimum; Max, maximum.

**Table 3 nutrients-12-02137-t003:** Associations between adherence to E-DII score and anthropometric and metabolic parameters.

E-DII Score							
Parameters	Tertile 1	Tertile 2		Tertile 3			
	−2.67 (−4.80–−1.75) ^1^	−1.05 (−1.71–−0.35) ^1^		0.97 (−0.34–3.23) ^1^			
	OR (95% CI)	OR (95% CI) ^2^ Tertile 2 vs. 1	*p*	OR (95% CI) ^2^ Tertile 3 vs. 1	*p*	OR (95% CI) ^2^ Tertiles 2 and 3 vs. 1	*p*
High BMI	-	1.297 (0.50–3.34)	0.59	1.260 (0.47–3.41)	0.65	1.280 (0.55–3.01)	0.57
High WC	-	1.089 (0.54–2.18)	0.81	1.574 (0.76–3.26)	0.22	1.283 (0.69–2.37)	0.43
High BP	-	3.029 (1.01–9.11)	**0.049**	1.135 (0.35–3.71)	0.83	1.928 (0.68–5.44)	0.21
High HOMA-IR	-	2.667 (1.11–6.43)	0.03	1.349 (0.50–3.68)	0.56	2.047 (0.89–4.70)	0.09
Low HDL-C	-	2.318 (1.04–5.16)	0.04	2.414 (1.04–5.58)	0.04	2.359 (1.13–4.92)	0.02
High LDL-C	-	1.200 (0.50–2.89)	0.68	1.183 (0.48–2.93)	0.72	1.192 (0.54–2.62)	0.66
High TG	-	0.937 (0.34–2.59)	0.90	1.658 (0.62–4.41)	0.31	1.240 (0.52–2.94)	0.63

^1^ Mean score (range); ^2^ Model adjusted for gender and survival time. Cut-off values are described in Appendix A. OR, odds ratio; CI, confidence interval; BMI, body mass index; WC, waist circumference; BP, blood pressure; HOMA-IR, homeostasis model assessment-insulin resistance; HDL-C, high-density lipoprotein-cholesterol; LDL-C, low density lipoprotein-cholesterol; TG, triglycerides.

**Table 4 nutrients-12-02137-t004:** Associations between ultra-processed foods and anthropometric and metabolic parameters.

% Ultra-Processed Foods According to the NOVA Classification							
Parameters	Tertile 1	Tertile 2		Tertile 3			
	34.2% (6.9–43.0%) ^1^	50.0% (43.7–56.5%) ^1^		69.3% (56.6–93.4%) ^1^			
	OR (95% CI)	OR (95% CI) ^2^ Tertile 2 vs. 1	*p*	OR (95% CI) ^2^ Tertile 3 vs. 1	*p*	OR (95% CI) ^2^ Tertiles 2 and 3 vs. 1	***p***
High BMI	-	0.360 (0.10–1.24)	0.11	0.929 (0.34–2.58)	0.89	0.619 (0.25–1.54)	0.30
High WC	-	0.622 (0.28–1.37)	0.24	0.968 (0.44–2.13)	0.94	0.772 (0.39–1.51)	0.45
High BP	-	0.781 (0.24–2.57)	0.68	1.078 (0.36–3.33)	0.89	0.934 (0.35–2.53)	0.89
High HOMA-IR	-	0.341 (0.11–1.03)	0.06	0.763 (0.30–1.97)	0.58	0.533 (0.23–1.23)	0.14
Low HDL-C	-	1.410 (0.55–3.64)	0.48	3.885 (1.54–9.80)	**0.004**	2.323 (1.02–5.28)	**0.04**
High LDL-C	-	0.407 (0.15–1.13)	0.09	0.728 (0.29–1.84)	0.50	0.556 (0.25–1.26)	0.16
High TG	-	2.998 (0.74–12.1)	0.12	5.434 (1.38–21.4)	**0.02**	4.021 (1.12–14.5)	**0.03**

^1^ Mean score (range); ^2^ Model adjusted for gender, survival time and energy intake. Cut-off values are described in Appendix A. OR, odds ratio; CI, confidence interval; BMI, body mass index; WC, waist circumference; BP, blood pressure; HOMA-IR, homeostasis model assessment-insulin resistance; HDL-C, high-density lipoprotein-cholesterol; LDL-C, low density lipoprotein-cholesterol; TG, triglycerides.

**Table 5 nutrients-12-02137-t005:** Association between adherence to nutritional scores and having ≥2 cardiometabolic complications.

Presence of 2 or More Cardiometabolic Complications							
Scores	OR (95% CI) Tertile 1	OR (95% CI) Tertile 2 vs. 1	*p*	OR (95% CI) Tertile 3 vs. 1	*p*	OR (95% CI) Tertiles 2 and 3 vs. 1	*p*
MEDAS ^2^	-	0.800 (0.33–1.97)	0.63	1.380 (0.54–3.50)	0.30	1.279 (0.58–2.80)	0.54
KIDMED ^2^	-	0.424 (0.12–1.57)	0.20	0.735 (0.23–2.40)	0.61	0.728 (0.25–2.13)	0.56
HDI-2018 ^2^	-	1.079 (0.52–2.23)	0.84	1.191 (0.58–2.43)	0.63	0.728 (0.25–2.13)	0.56
HEI-2015 ^3^	-	1.053 (0.52–2.12)	0.88	0.750 (0.36–1.55)	0.44	0.911 (0.49–1.68)	0.77
E-DII ^3^	-	2.506 (1.22–5.15)	**0.01**	1.613 (0.74–3.50)	0.23	2.076 (1.07–4.07)	**0.03**
FRAP ^2^	-	1.509 (0.73–3.13)	0.27	1.245 (0.57–2.73)	0.58	1.391 (0.71–2.71)	0.33
% UPF ^2,4^	-	0.647 (0.29–1.47)	0.30	1.128 (0.51–2.49)	0.77	0.856 (0.43–1.70)	0.66

Presence of at least two of these four factors: obesity, pre-hypertension/hypertension, insulin resistance and dyslipidemia; ^2^ Model adjusted for gender, survival time and energy intake; ^3^ Model adjusted for gender and survival time; ^4^ %UPF, percentage of contribution of ultra-processed foods to total dietary energy intake using NOVA classification. OR, odds ratio; CI, confidence interval.

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
