# Peer review of "Diet Quality Is Associated with Cardiometabolic Outcomes in Survivors of Childhood Leukemia"

_nutrients, 2020, doi:10.3390/nu12072137_

Round 1

Reviewer 1 Report

The authors describe a comprehensive evaluation of childhood leukemia survivors dietary habits and their association with cardiac and metabolic health indices.  Based on patients in the PETALE study, they reviewed 241 patients, determined their dietary habits, and measured multiple health outcomes.  They performed associations with the available data, and found, as expected, that cancer survivors generally had poor dietary habits, and adhered poorly to recommended diets.  They found multiple associations with poor diet and elevated cholesterol and triglycerides, as well as poor diet and hypertension, and poor diet and elevations in certain cytokines.

Overall, the paper is very well written, thorough, and relatively easy to follow for the readers.  Strengths include a very well written introduction, as well as a very large amount of descriptive data (both in the paper and supplementary info), which is helpful to the reader.  Diet in cancer survivors is a field with a relatively small amount of data, and it is a field worth further study.

Minor critiques:

It may be helpful to the reader to have slightly more information about the cancer survivors themselves.  Specifically, knowing the number of patients who received intensive therapies associated with poor long term cardiac outcomes: what number of patients received anthracyclines?  radiation?  bone marrow transplant?  Do they authors have this data?  This could be added to Table 1.

Intro and Methods are very thorough.

The units used for glucose, insulin, cholesterol, and triglycerides are based on Canadian/European system and difficult for US readers.  Could consider a column using US measurements (ex Chol mg/dL).  Or this could be in supplementary info.

Discussion:

The authors allude to this, but how do the dietary data compare to the general population?  Do cancer survivors have a worse diet than the general population?  Or is it just equally bad?

The authors could mention poor compliance and its association with the AYA population (adolescent and young adult).  Much of the patient population would fall into the AYA group, and this study is important in studying that understudied group.  Maybe a few brief sentences in the discussion on AYA and compliance.

Limitations are well described.  As above, a few more comparisons to the general population would be helpful.  I actually think the number of survivors studied is quite robust.

Overall very well written and an important addition to the literature.

Author Response

General comment:

The authors describe a comprehensive evaluation of childhood leukemia survivors dietary habits and their association with cardiac and metabolic health indices.  Based on patients in the PETALE study, they reviewed 241 patients, determined their dietary habits, and measured multiple health outcomes.  They performed associations with the available data, and found, as expected, that cancer survivors generally had poor dietary habits, and adhered poorly to recommended diets.  They found multiple associations with poor diet and elevated cholesterol and triglycerides, as well as poor diet and hypertension, and poor diet and elevations in certain cytokines.

Overall, the paper is very well written, thorough, and relatively easy to follow for the readers.  Strengths include a very well written introduction, as well as a very large amount of descriptive data (both in the paper and supplementary info), which is helpful to the reader.  Diet in cancer survivors is a field with a relatively small amount of data, and it is a field worth further study

Authors’ Response: We thank the reviewer for his/her positive comments regarding our manuscript. He/she will find below the point-by-point answers.

Minor critiques:

It may be helpful to the reader to have slightly more information about the cancer survivors themselves.  Specifically, knowing the number of patients who received intensive therapies associated with poor long term cardiac outcomes: what number of patients received anthracyclines?  radiation?  bone marrow transplant?  Do they authors have this data?  This could be added to Table 1.

Authors’ Response :

We thank the Reviewer for this comment. We agree that adding information on the cohort description can be useful for the reader. We have added information about the cohort in the Materials and Methods section and in Table 1. Also, patients who had received bone marrow transplant were not eligible to the PETALE study. This information was added to the Methods section.

Materials and Methods

2.1 Study population 

- A detailed presentation of the PETALE study and cohort is provided in Marcoux el al. [36]. (Lines 76-77)

- Event-free patients who had not suffered from refractory ALL, relapsed, or received hematopoietic stem cell transplant 5 years or more after diagnosis were eligible. (Lines 81-84)

Table 1

Data on exposure to CRT exposure were added to Table 1. 

Reviewer’s comment: Intro and Methods are very thorough

Authors’ Response: We thank the Reviewer for this positive comment.

Reviewer’s comment: The units used for glucose, insulin, cholesterol, and triglycerides are based on Canadian/European system and difficult for US readers.  Could consider a column using US measurements (ex Chol mg/dL).  Or this could be in supplementary info.

Authors’ Response: As a matter of fact, data are presented in the International System of units. To facilitate the reading, we have added the units in the Imperial system when appropriate (please refer to Supplementary Tables 1, 7 and 8).

Reviewer’s comment:

The authors allude to this, but how do the dietary data compare to the general population?  Do cancer survivors have a worse diet than the general population?  Or is it just equally bad.

Authors’ Response: Unfortunately, as stated in the study limitations, our study did not include a non-cancer control group. Thereby, it is not possible for us to compare diet quality with the general population. Our instinct tells us that the diet of our population of cALL survivors is similar to the general population, but this cannot be demonstrated with our data.

We have addressed this to the Discussion section (page 12): line 470

The lack of a non-cancer control group is another study limitation that prevents comparisons with the general population.

Reviewer’s comment:

The authors could mention poor compliance and its association with the AYA population (adolescent and young adult).  Much of the patient population would fall into the AYA group, and this study is important in studying that understudied group.  Maybe a few brief sentences in the discussion on AYA and compliance.

Authors’ Response: This is a very interesting point and we thank the Reviewer to bring it to our attention. Text was added in the Discussion section (lines 453-460) accordingly:

The PETALE cohort is mostly comprised of AYA cALL survivors, a population that is understudied and that faces particular challenges in terms of compliance to treatment and ability to follow dietary recommendations. In particular, low consumption of fruits and vegetables, fibers and high intakes of high-fat foods have been reported in AYA cancer survivors. Besides, high intakes of refined grains, empty calories, fruit juice and food high in sugar and fat were noted in children and adolescents from the general population. Therefore, it is of high interest to study dietary and lifestyle habits in the very specific population of AYA cancer survivors. 

Reviewer’s comment:

Limitations are well described.  As above, a few more comparisons to the general population would be helpful.  I actually think the number of survivors studied is quite robust.

Overall very well written and an important addition to the literature.

Authors’ Response: We thank the reviewer for his/her positive comments.

Reviewer 2 Report

I read with interest in the manuscript. Behrard et other authors write on a very important item on vulnerable patients. I consider this paper at a high level. I appreciate the research question and evaluate it well wrote
Only some suggestions
1. Introduction: well written, if you can explain the data on the burden of childhood acute lymphoblastic leukemia (cALL) with global data and data from your country
2. Methods: well wrote
3. results: I appreciate a lot also table and figure
4. Discussion: well done, if you can use the terms of children at risk to indicate all children in all part of the world vulnerable (oncologic , under nutrition, et, cite this paper The At Risk Child Clinic (ARCC): 3 Years of Health Activities in Support of the Most Vulnerable Children in Beira, Mozambique. Int J Environ Res Public Health. 2018;15(7):1350. Published 2018
Consider also in your discussion Italian data (ex cite : Childhood and Adolescence Cancers in the Palermo Province (Southern Italy): Ten Years (2003⁻2012) of Epidemiological Surveillance. Int J Environ Res Public Health. 2018;15(7):1344. Published 2018 Jun 26)

Author Response

I read with interest in the manuscript. Behrard et other authors write on a very important item on vulnerable patients. I consider this paper at a high level. I appreciate the research question and evaluate it well wrote. Only some suggestions

Authors’ Response: We thank the reviewer for his/her positive comments regarding our manuscript. He/she will find below the point-by-point answers.

Reviewer’s comment: Introduction: well written, if you can explain the data on the burden of childhood acute lymphoblastic leukemia (cALL) with global data and data from your country.

Authors’ Response: Global and Canadian data on cALL and survivors were added in the Introduction section : Lines 38-43

Pediatric cancers are the leading causes of disease-related mortality in Canadian children, and acute lymphoblastic leukemia (ALL) is the most common of these cancers. An average of 880 Canadian children under the age of 15 are diagnosed with cancer each year and the incidence of pediatric cancers has remained relatively stable since 1992. Leukemia accounts for 32% of all pediatric cancers. In recent years, therapeutics advances have achieved a survival rate for childhood ALL (cALL) that exceeds 90%. Therefore, the population of survivors is growing in Canada and other parts of the world.

Reviewer’s comment: Methods: well wrote and Results: I appreciate a lot also table and figure.

Authors’ Response: We thank the reviewer for his/her positive comments.

Reviewer’s comment: Discussion: well done, if you can use the terms of children at risk to indicate all children in all part of the world vulnerable (oncologic, under nutrition, et, cite this paper The At Risk Child Clinic (ARCC): 3 Years of Health Activities in Support of the Most Vulnerable Children in Beira, Mozambique. Int J Environ Res Public Health. 2018;15(7):1350. Published 2018)

Authors’ Response: We thank the reviewer for this suggestion. We read this paper with great interest. However, we do not see how this particular concept could be integrated to our work on survivors of childhood cancer at risk of cardiometabolic complications. We would gladly integrate this information if the Reviewer would kindly guide us on where and how to do so.

Consider also in your discussion Italian data (ex cite : Childhood and Adolescence Cancers in the Palermo Province (Southern Italy): Ten Years (2003⁻2012) of Epidemiological Surveillance. Int J Environ Res Public Health. 2018;15(7):1344. Published 2018 Jun 26).

Authors’ Response: We thank the reviewer for his/her suggestions. This paper was added to the Introduction section: (Lines 38-43).

Pediatric cancers are the leading causes of disease-related mortality in Canadian children, and acute lymphoblastic leukemia (ALL) is the most common of these cancers. An average of 880 Canadian children under the age of 15 are diagnosed with cancer each year and the incidence of pediatric cancers has remained relatively stable since 1992. Leukemia accounts for 32% of all pediatric cancers. In recent years, therapeutics advances have achieved a survival rate for childhood ALL (cALL) that exceeds 90%. Therefore, the population of survivors is growing in Canada and other parts of the world. Unfortunately, cALL survivors are at greater risk of developing long-term chronic health complications.